# Regeneration of Non-Alcoholic Fatty Liver Cells Using Chimeric FGF21/HGFR: A Novel Therapeutic Approach

**DOI:** 10.3390/ijms25063092

**Published:** 2024-03-07

**Authors:** Sung-Jun Kim, So-Jung Kim, Jeongeun Hyun, Hae-Won Kim, Jun-Hyeog Jang

**Affiliations:** 1Department of Biochemistry, Inha University School of Medicine, Incheon 22212, Republic of Korea; tjdwns9617@naver.com; 2Institute of Tissue Regeneration Engineering (ITREN), Dankook University, Cheonan 31116, Republic of Korea; r00by8340@naver.com (S.-J.K.); j.hyun@dankook.ac.kr (J.H.); kimhw@dku.edu (H.-W.K.); 3Department of Nanobiomedical Science and BK21 PLUS NBM Global Research Center for Regenerative Medicine, Dankook University, Cheonan 31116, Republic of Korea; 4Mechanobiology Dental Medicine Research Center, Dankook University, Cheonan 31116, Republic of Korea; 5College of Dentistry, Dankook University, Cheonan 31116, Republic of Korea; 6UCL Eastman-Korea Dental Medicine Innovation Centre, Dankook University, Cheonan 31116, Republic of Korea; 7Cell & Matter Institute, Dankook University, Cheonan 31116, Republic of Korea

**Keywords:** alpha mouse liver 12 (AML12), fibroblast growth factor 21 (FGF21), hepatocyte growth factor receptor (HGFR), chimeric FGF21/HGFR, liver regeneration

## Abstract

Non-alcoholic fatty liver disease (NAFLD) has emerged as a significant liver ailment attributed to factors like obesity and diabetes. While ongoing research explores treatments for NAFLD, further investigation is imperative to address this escalating health concern. NAFLD manifests as hepatic steatosis, precipitating insulin resistance and metabolic syndrome. This study aims to validate the regenerative potential of chimeric fibroblast growth factor 21 (FGF21) and Hepatocyte Growth Factor Receptor (HGFR) in NAFLD-afflicted liver cells. AML12, a murine hepatocyte cell line, was utilized to gauge the regenerative effects of chimeric FGF21/HGFR expression. Polysaccharide accumulation was affirmed through Periodic acid–Schiff (PAS) staining, while LDL uptake was microscopically observed with labeled LDL. The expression of FGF21/HGFR and NAFLD markers was analyzed by mRNA analysis with RT-PCR, which showed a decreased expression in acetyl-CoA carboxylase 1 (*ACC1*) and sterol regulatory element binding protein (*SREBP*) cleavage-activating protein (*SCAP*) with increased expression of hepatocellular growth factor (*HGF*), hepatocellular nuclear factor 4 alpha (*HNF4A*), and albumin (*ALB*). These findings affirm the hepato-regenerative properties of chimeric FGF21/HGFR within AML12 cells, opening novel avenues for therapeutic exploration in NAFLD.

## 1. Introduction

Non-alcoholic fatty liver disease (NAFLD) has become one of the leading causes of chronic liver disease globally, with its prevalence reaching epidemic proportions [1,2]. NAFLD encompasses a spectrum of liver pathologies, ranging from simple hepatic steatosis, characterized by excessive fat accumulation in hepatocytes, to non-alcoholic steatohepatitis (NASH), which involves inflammation, hepatocyte injury, and fibrosis [3]. The increasing prevalence of NAFLD is closely associated with obesity, insulin resistance, metabolic syndrome, and sedentary lifestyles [4]. Consequently, the demand for therapeutic approaches for NAFLD and its progressive forms has never been greater.

In NAFLD, the liver is affected by various metabolic problems compared to a normal liver [5]. The problem is that the fatty liver caused by NAFLD has insulin resistance properties, so insulin cannot affect liver cells, which causes problems with blood sugar control and fat absorption, as well as protein production and breakdown in liver cells [6]. In addition to these problems, fatty liver cells in NAFLD have a reduced ability to regenerate into normal liver cells, which can lead to other liver diseases. If the condition becomes more severe, it can lead to liver cancer [7]. Because NAFLD is a disease that occurs in the liver that can lead to serious consequences over time, research into treatments to regenerate the liver is needed before this process progresses.

The liver’s regenerative capacity is a vital defense mechanism that allows it to respond to injuries and maintain functional integrity [8]. However, in the context of NAFLD, the regenerative capacity of hepatocytes is often compromised, leading to impaired liver regeneration and potential progression to more severe liver diseases [9]. Therefore, developing innovative strategies to enhance liver regeneration in NAFLD patients has become a significant research focus in hepatology and regenerative medicine.

Fibroblast growth factor 21 (FGF21) and Hepatocyte Growth Factor Receptor (HGFR, also known as c-Met) have emerged as promising therapeutic targets in liver regeneration research [10,11]. FGF21, a member of the fibroblast growth factor superfamily, has shown potent metabolic regulatory effects, including its ability to improve lipid metabolism and insulin sensitivity [12]. Importantly, recent studies have unveiled additional roles for FGF21 in tissue repair and regeneration beyond its metabolic functions [13]. HGFR, a receptor tyrosine kinase, is a critical player in hepatocyte proliferation, migration, and tissue repair processes during liver regeneration [14].

Several chimeric growth factor receptors have shown success in their protein activity, achieved by creating hybrid proteins through the synthesis of growth factor receptors and other proteins [15]. Based on the above, HGFR is known to affect hepatocyte regeneration, and FGF21 is also known to regenerate hepatocytes, so we conducted this study to question whether the proteins expressed by chimeric processes could activate hepatocyte regeneration.

The concept of chimeric fusion proteins combining FGF21 and HGFR has introduced a novel therapeutic approach to harness the regenerative potential of these two molecules in a synergistic manner. The chimeric FGF21/HGFR fusion protein holds the promise of addressing both the metabolic imbalances associated with NAFLD and the impaired liver regeneration observed in these patients.

The objective of this thesis is to investigate and confirm the regeneration of non-alcoholic fatty liver cells using chimeric FGF21/HGFR as a novel therapeutic approach. By exploring the molecular mechanisms underlying liver regeneration, the specific roles of FGF21 and HGFR in tissue repair, and their potential synergistic effects as a chimeric fusion protein, we aim to shed light on its efficacy and translational potential as a therapeutic intervention for NAFLD.

## 2. Results

### 2.1. Construction of Stable AML12 Cell Line for Expression of Chimeric FGF21/HGFR

In order to study the functionality of chimeric FGF21/HGFR, it is necessary to insert the vector into cells and express genes. For the expression of chimeric FGF21/HGFR in AML12, the sequence encoding FGF21/HGFR was cloned into the pcDNA3.1 expression vector. This vector was designed to include the GFP gene to confirm expression (Figure 1).

### 2.2. FGF21/HGFR on AML12 Cells Are Closely Related Glycogen Accumulation Ability

AML12 (alpha mouse liver 12) cells are hepatocytes isolated from the normal liver. To confirm the recovery of hepatocyte function in AML12, the cells were transfected with chimeric FGF21/HGFR, and Periodic acid–Schiff (PAS) staining confirmed that glycogen was accumulated in AML12-expressing chimeric FGF21/HGFR. In order to confirm glycogen expression in AML12 transfected with chimeric FGF21/HGFR, Schiff’s reagent was used to stain the glycogen inside the cells, and cell nuclei were stained using Mayer’s Hematoxylin. In AML12 cells expressing chimeric FGF21/HGFR, it was confirmed that magenta-stained glycogen was produced, and in the control group, glycogen was not observed, as shown in Figure 2. RT-PCR analysis confirmed the increased expression of chimeric FGF21/HGFR in FGF21/HGFR-transfected AML12 cells compared to the controls (Figure 2E).

### 2.3. FGF21/HGFR Promotes LDL Uptake on AML12 Cells

This study confirmed the liver regeneration ability of chimeric FGF21/HGFR at the cell level through LDL uptake and regulation. AML12 was transformed with chimeric FGF21/HGFR and confirmed through fluorescence microscopy. To observe chimeric FGF21/HGFR-induced lipid accumulation and lipid uptake, AML12 cells treated with 10 μg/mL of labeled LDL for 3 h were observed under a fluorescence microscope. We confirmed that AML12 uptakes more LDL in the FGF21/HGFR group compared to the control group, and these data suggest that the function of hepatocytes in AML12 was enhanced (Figure 3).

### 2.4. Evaluation of the Expression for Hepatocyte-Function-Related Genes

The induction of NAFLD is closely related to free fatty acids, a form of saturated fatty acids, of which palmitate is an important component. RT-PCR was performed to confirm regeneration at the mRNA level when chimeric FGF21/HGFR was involved in AML12 by treating palmitate. The results showed a significant decrease in Acetyl-CoA carboxylase 1 (*Acc1*) (Figure 4A) and sterol regulatory element binding protein (SREBP) cleavage activating protein (*Scap*) at the mRNA level in AML12 transfected with FGF21/HGFR compared to the control group in the NAFLD state, and an upregulation of hepatocyte growth factor (*Hgf*), and *Albumin* (Figure 4B). These data indicate that chimeric FGF21/HGFR may be helpful in studying hepatocyte proliferation and liver regeneration mechanisms in NAFLD.

## 3. Discussion

The liver, a highly metabolic organ, possesses remarkable regenerative capabilities. However, the surge in obesity rates has led to an alarming increase in non-alcoholic fatty liver disease (NAFLD) [16]. NAFLD’s impact on liver regenerative capacity poses a pressing concern, prompting our investigation into the regenerative potential of chimeric fibroblast growth factor 21 (FGF21)/Hepatocyte Growth Factor Receptor (HGFR) within the context of NAFLD. In previous studies, FGF21 has attracted attention as a new target for NAFLD [17]. The deletion of HGFR in hepatocyte causes NASH [18], and the relationship between NASH and NAFLD has been studied continuously [19]. This study aims to unravel its effects on hepatocyte function and gene expression, with the ultimate goal of shedding light on novel avenues for NAFLD treatment.

Before delving into the core experiments, we meticulously confirmed the expression of chimeric FGF21/HGFR within AML12 cells. Although RT-PCR analysis of mRNA expression may not guarantee the proper expression of FGF21/HGFR fusion protein and requires further validation, our results indicate that the insertion of a GFP tag into pcDNA3.1 and the subsequent fluorescent gene expression in AML12 cells enabled us to visually affirm the expression of chimeric FGF21/HGFR. This preliminary step ensured that subsequent experiments were conducted exclusively on cells displaying fluorescence, establishing a robust foundation for our investigation.

Glycogen synthesis serves as a vital marker of hepatocyte function, with profound implications for liver regeneration [20,21]. Our investigation, employing the PAS staining assay, revealed a striking contrast between the control and chimeric FGF21/HGFR-expressing cells. The notable accumulation of glycogen in the latter signifies an enhanced capacity for glucose conversion, pointing toward substantial functional improvement induced by chimeric FGF21/HGFR.

Effective LDL uptake stands as a pivotal indicator of hepatocyte function [22]. Our findings in the LDL uptake assay corroborate the hypothesis of functional enhancement attributed to liver regeneration. In contrast to control cells, those expressing chimeric FGF21/HGFR exhibited a substantial increase in LDL uptake. This heightened capacity for lipoprotein metabolism underscores the potential of chimeric FGF21/HGFR to alleviate the lipid dysregulation associated with NAFLD.

To comprehensively characterize liver regeneration and growth in AML12 cells under the influence of chimeric FGF21/HGFR, we scrutinized the mRNA levels of key markers. These included Acc1 and Scap, associated with lipid metabolism, whose expressions were notably influenced by both NAFLD induction and chimeric FGF21/HGFR expression. Furthermore, our analysis unveiled an upregulation of hepatocyte growth factor (HGF) and albumin, suggesting enhanced hepatocyte functionality and liver regeneration potential, in concurrence with previous NAFLD research findings.

The findings of this study align closely with existing NAFLD research, bolstering the notion that chimeric FGF21/HGFR holds promise in improving liver regeneration function in NAFLD-afflicted AML12 cells [23,24,25,26]. These observations not only provide a deeper understanding of NAFLD but also open doors to novel perspectives in liver regeneration treatment.

While our current study utilized AML12 cells, which are a murine hepatocyte cell line, we recognize the importance of investigating the therapeutic potential of chimeric FGF21/HGFR in human liver cells. Including experiments with a human liver cell line would provide valuable insights into the potential applicability of our findings to clinical scenarios. Human liver cell lines, such as HepG2 or Huh7, could offer a more direct representation of human hepatic physiology and response to treatment [27]. We acknowledge the significance of moving towards more clinically relevant models, and we are considering the inclusion of experiments using human liver cell lines in our future research. This step would contribute to a more comprehensive understanding of the regenerative properties of chimeric FGF21/HGFR and its potential as a therapeutic intervention for non-alcoholic fatty liver disease in humans.

Indeed, exploring the specific overexpression of chimeric FGF21/HGFR in a mouse model following partial hepatectomy or acute liver injury would be a valuable avenue for extending the applicability of our findings. This approach would allow us to assess the regenerative potential of chimeric FGF21/HGFR in a more physiologically relevant context, simulating conditions of liver regeneration and injury. The use of a mouse model with partial hepatectomy or acute liver injury could provide valuable insights into the therapeutic efficacy and mechanisms underlying the regenerative effects of chimeric FGF21/HGFR. It would allow us to observe its impact on liver regeneration, tissue repair, and functional recovery in a dynamic environment that mimics the challenges faced by the liver in clinical scenarios. We acknowledge the importance of translational research and plan to consider such in vivo studies in future phases of our investigation. This step would further contribute to a comprehensive understanding of the therapeutic potential of chimeric FGF21/HGFR in the context of non-alcoholic fatty liver disease.

In conclusion, this study underscores the regenerative potential of chimeric FGF21/HGFR in the context of NAFLD, offering promising prospects for therapeutic interventions. By enhancing hepatocyte function and influencing gene expression patterns, chimeric FGF21/HGFR emerges as a potential candidate for further exploration in the quest to address NAFLD and advance liver regeneration research.

## 4. Materials and Methods

### 4.1. Construction of Chimeric FGF21/HGFR Receptor

To generate the chimeric FGF21/HGFR receptor construct, a synthetic HGFR sequence encompassing the intracellular domain and transmembrane domain from amino acid (aa) 251 to 712 of HGFR including a signal peptide sequence was synthesized (Genotech, Daejeon, Republic of Korea). This sequence was then inserted into the pCDNA3.1 expression vector (Invitrogen, Carlsbad, CA, USA) using the restriction enzymes NheI and XhoI. In addition, a synthetic FGF21 sequence, encoding the full length of the protein from aa 28 to 208 of FGF21 including green fluorescent protein GFP (238 aa) produced by Genotech (Daejeon, Republic of Korea) was fused to the transmembrane domain of HGFR using the restriction enzymes BglII and BamHI.

### 4.2. Cell Culture and Treatment

AML12 cells (ATCC, Manassas, VA, USA) were cultured in Dulbecco Modified Eagle Medium/F12 (DMEM/F12) (Gibco, Carlsbad, CA, USA) containing 10% (*v*/*v*) heat-inactivated fetal bovine serum (Corning, Corning, NY, USA, 35-015-CV), 1% insulin–transferrin–selenium 100× (Gibco, #41400-045), and 40 ng/mL dexamethasone (Sigma-Aldrich, St Louis, MO, USA) at 37 °C in a humidified atmosphere with 5% CO_2_. Cells were treated for 24 h with 250 µM palmitate (Cyaman, Ann Arbor, MI, USA).

For transfection, AML12 cells were seeded in a 6-well plate (SPL, Pocheon, Republic of Korea) at 8 × 10^4^ cells/mL overnight. After that, they were transfected with chimeric FGF21/HGFR DNA and control DNA for 48 h. This transfection assay was performed according to the Lipofectamine 3000 (Invitrogen, #L3000001) protocol. All further assays were performed within 24 h after transfection.

### 4.3. Periodic Acid–Schiff (PAS) Staining

In the control group and FGF21 group (transfected chimeric FGF21/HGFR), the AML12 cells were seeded at a density of 3 × 10^5^ cells/well and incubated overnight at 37 °C. The medium was removed from the plate, cells were rinsed with PBS two times and fixed with 4% paraformaldehyde (Tech-innovation, Chuncheon, Republic of Korea) for 10 min at room temperature, and then washed two times with PBS, oxidized for 5 min with 1% periodic acid, and washed two times with PBS. Cells were then stained with Schiff’s reagent (Abcam, Cambridge, UK) for 15 min, washed 2 times with PBS, then stained with Mayer’s Hematoxylin for 2 min and washed 3 times with PBS before microscopic examination.

### 4.4. LDL Uptake Assay

This study was conducted using Image-iT™ Low Density Lipoprotein Uptake Kit, pHrodo™ Red (Invitrogen, #I34360). AML12 cells were seeded to 30–40% confluency in a confocal dish (SPL, #211350) for 24 h. A serum starvation medium consisting of 10 mL of FluoroBrite DMEM (Gibco, A1896701) and 200 mg of 2% bovine serum albumin (BSA) (Sigma-Aldrich, St Louis, MO, USA) was additionally added to the dish and incubated for 14 h. An amount of 10 μg/mL of labeled LDL was added to the dish and incubated for 3 h. Hoechst was added and incubated for 10 min to stain the nuclei, followed by PBS washing. Stained cells were observed under a microscope and fluorescence was observed.

### 4.5. Reverse Transcription (RT)-PCR and Real-Time PCR

Total RNA was extracted using Ribospin™ (Geneall, Seoul, Republic of Korea) and cDNA was synthesized (Applied Biosystems, #4368814). mRNA expression of genes (*Acc1*, *Scap*, *Hnf4a*, *Hgf*, *Albumin*, and *Hgfr*) was examined by quantitative real-time PCR. The primer sequences of these genes used in this experiment are listed in Table 1. All real-time PCR experiments were analyzed on an ABI Step One real-time PCR system (Applied Biosystem, Foster City, CA, USA). Each real-time PCR reaction was performed in a 20 μL reaction mixture containing 0.1 μM of each primer, 10 μL of SYBR Hi-ROX (Bioline, Memphis, TN, USA), and 5 µL of template cDNA. The real-time PCR cycles were as follows: 1 cycle at 95 °C for 10 min (enzyme activation); 40 cycles at 95 °C for 15 s (denaturation), 60 °C for 15 s (annealing), and 72 °C for 30 s (extension); and 1 cycle at 95 °C for 15 s, 60 °C for 1 min, and 95 °C for 15 s (melt curve stage).

### 4.6. Statistical Analysis

Experiments were performed three times each in triplicate. The results are expressed as mean ± standard deviation. Statistical analysis was performed using Student’s *t*-test. (* *p* < 0.05, ** *p* < 0.01, and *** *p* < 0.001).

## 5. Conclusions

In this study, we investigated the regenerative potential of chimeric FGF21/HGFR as a therapeutic strategy for non-alcoholic fatty liver disease (NAFLD). Through cell transfection and analysis, we observed that chimeric FGF21/HGFR has the effect of regenerating normal hepatocellular properties in non-alcoholic fatty liver. These findings suggest chimeric FGF21/HGFR’s potential as a therapeutic intervention for NAFLD.

## Figures and Tables

**Figure 1 ijms-25-03092-f001:**
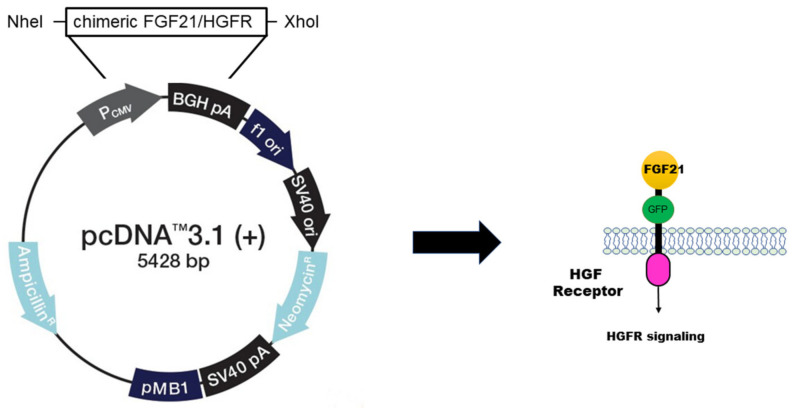
Construction of chimeric FGF21/HGFR receptor. For the expression of chimeric FGF21/HGFR in AML12, the sequence encoding FGF21/HGFR was cloned into the pcDNA3.1 expression vector.

**Figure 2 ijms-25-03092-f002:**
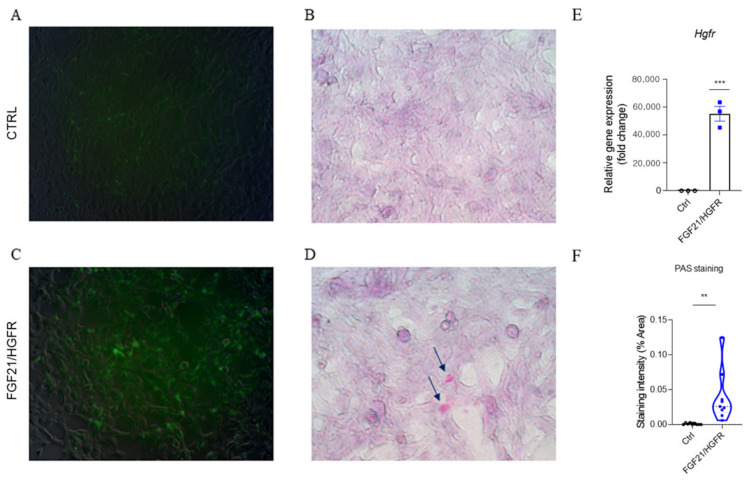
Glycogen accumulation capacity was increased in chimeric FGF21/HGFR on AML12: (**A**,**C**) The GFP gene was expressed in AML12 and observed by fluorescence. (**B**,**D**) Accumulation of glycogen was observed through a microscope (glycogen accumulation, arrows), (magnification, ×200). The mRNA levels of *Hgfr* were analyzed by RT-PCR (**E**). Quantification of PAS staining intensity (**F**). ** *p* < 0.01; *** *p* < 0.001.

**Figure 3 ijms-25-03092-f003:**
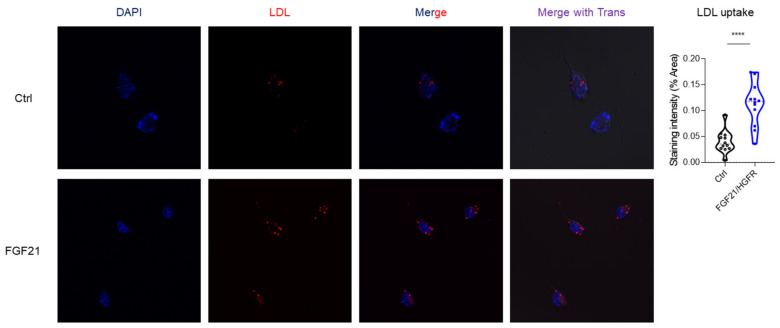
LDL uptake was upregulated in chimeric FGF21/HGFR on AML12. The AML12 cells with labeled LDL were detected by immunofluorescence staining, (magnification, ×200). **** *p* < 0.0001.

**Figure 4 ijms-25-03092-f004:**
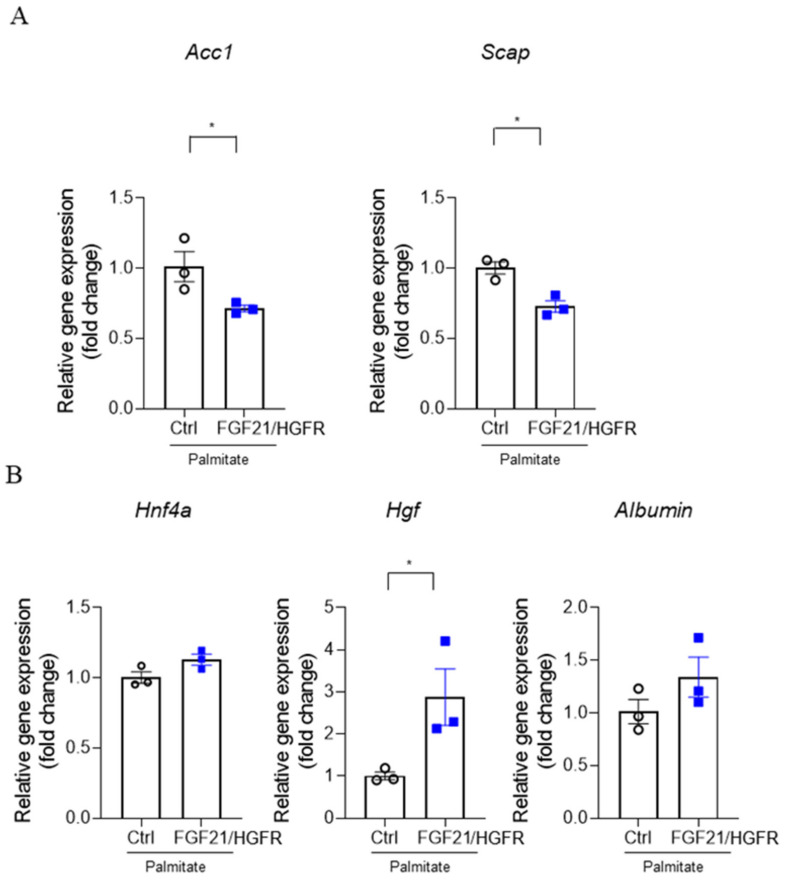
In AML12 where steatosis was induced by palmitate, chimeric FGF21/HGFR showed an indicator of liver regeneration: (**A**) The mRNA levels of *Acc1* and *Scap* were analyzed by real-time PCR, and the comparative mRNA levels of each gene were assessed relative to the mRNA levels of the housekeeping gene, *Gapdh*. (**B**) The mRNA levels of *Hnf4a*, *Hgf*, and *Albumin* were analyzed by real-time PCR, and the comparative mRNA levels of each gene were assessed relative to the mRNA levels of the housekeeping gene, *Gapdh*. * *p* < 0.05.

**Table 1 ijms-25-03092-t001:** Sequence of primers used for quantitative real-time PCR.

Genes	Forward Primer	Reverse Primer
*GAPDH*	5′-CTGCACCACCAACTGCTTAG-3′	5′-GTCTTCTGGGTGGCAGTGAT-3′
*Hnf4a*	5′-TGCGAACTCCTTCTGGATGACC-3′	5′-CAGCACGTCCTTAAACACCATGG-3′
*Albumin*	5′-CAGTGTTGTGCAGAGGCTGACA-3′	5′-GGAGCACTTCATTCTCTGACGG-3′
*Scap*	5′-AGAATTCCACAGGTCCCGTT-3′	5′-CTGCGCATCCTATCCAATTC-3′
*Acc1*	5′-TGACAGACTGATCGCAGAGAAAG-3′	5′-TGGAGAGCCCCACACACA-3′
*Hgf*	5′-GTCCTGAAGGCTCAGACTTGGT-3′	5′-CCAGCCGTAAATACTGCAAGTGG-3′
*Hgfr*	5′-GCAATTTCTTCAACCGTCCTTG-3′	5′-AAACCATTGGACAAAGTGTG-3′

## Data Availability

Data is contained within the article.

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
