# Peer review of "Regeneration of Non-Alcoholic Fatty Liver Cells Using Chimeric FGF21/HGFR: A Novel Therapeutic Approach"

_ijms, 2024, doi:10.3390/ijms25063092_

Round 1

Reviewer 1 Report

Comments and Suggestions for Authors

The manuscript submitted by Kim et al. focuses on the impact of chimeric FGF21/HGFR expression on non-alcoholic liver regeneration. Unfortunately the presented data is extremely preliminary and does not show any significance. The FGF21/HGFR is generated without even stating why this chimera is selected and why the author believes that it can improve the status/regeneration of steatotic liver. Then this chimera is expressed in one cell line only, by generating stable cell line. The expression is not even clear in the provided low quality image. Glycogen storage is showed by PAS staining in two cells which are most likely artefact of the staining. Author states that glycogen storage shows the functionality of hepatocytes which is not correct. LDL uptake is shown in one vs three cells and is considered “large amount of LDL uptake” (line 104). in vitro gene expression of some genes are shown, which by the author are considered key NAFLD markers (n=3??). Interestingly, in the abstract the author claimed that Hnf4a and Alb are increased which based on the presented data is not true.

Comments on the Quality of English Language

N/A

Author Response

Dear. Editors and reviewers

Thank you for your kind letter, with regard to our manuscript of ijms-2800797 together with comments. We are thankful to you for the very valuable suggestions through the whole manuscript. We tried to revise the manuscript as much as possible in line with suggestion made by the editor and reviewers. Also, we highlighted using the "Track Changes" function in Microsoft Word. We are herewith enclosing improved manuscript. Thank you again for your kind considerations.

We hope the improved version will be acceptable for publication in “IJMS”.

Thank you.

Yours sincerely,

Jun-Hyeog Jang Ph.D. (Corresponding author)

Professor

Department of Biochemistry,

Inha University School of Medicine,

Michuhol-gu, Incheon 22212, Korea

Tel: +82-32-860-9863; Fax: +82-32-885-8302. E-mail: juhjang@inha.ac.kr

Our answers to reviewer’ suggestion is as follow:

# Reviewer’s Comments

  1. The manuscript submitted by Kim et al. focuses on the impact of chimeric FGF21/HGFR expression on non-alcoholic liver regeneration. Unfortunately the presented data is extremely preliminary and does not show any significance. The FGF21/HGFR is generated without even stating why this chimera is selected and why the author believes that it can improve the status/regeneration of steatotic liver. Then this chimera is expressed in one cell line only, by generating stable cell line. The expression is not even clear in the provided low quality image. Glycogen storage is showed by PAS staining in two cells which are most likely artefact of the staining. Author states that glycogen storage shows the functionality of hepatocytes which is not correct. LDL uptake is shown in one vs three cells and is considered “large amount of LDL uptake” (line 104). in vitro gene expression of some genes are shown, which by the author are considered key NAFLD markers (n=3??). Interestingly, in the abstract the author claimed that Hnf4a and Alb are increased which based on the presented data is not true.

Answer:

We appreciate your thoughtful and constructive feedback on our manuscript. We understand and share your commitment to improving the quality and significance of scientific research. We have carefully considered your comments, and we would like to address each concern systematically. The reason why chimeric FGF21/HGFR was selected for NAFLD treatment in this paper is that HGFR is known to have an effect on hepatocyte regeneration, and FGF21 is also known to regenerate hepatocyte, so I started with the question that expressing protein by chimeric could activate hepatocyte regeneration. The part that was lacking in explanation about this content was additionally written in the introduction part(PAGE 2). This paper was conducted in one cell line, but the follow-up experiment was written in the Disscusion part (PAGE 6).

We have employed clearer images to illustrate the expression of GFP in control- or FGF21/HGFR-transfected cells.

In recent studies, it has been reported that glycogen storage shows the functionality of hepatocytes(Soon GST.; Torbenson M.; The Liver and Glycogen: In Sickness and in Health. Int J Mol Sci. 2023 Mar 24;24(7):6133). We stated that we were comparing the control group’s LDL uptake ability with the FGF21/HGFR group’s LDL uptake ability, and as suggested, we excluded the phrase “largen mount of LDL uptake” (PAGE 4). In Figure 4, several markers can be identified on the in vitro, and this part is shown in the Disccusion section (PAGE 6). we recognize that the way this was articulated may have led to misunderstanding. Accordingly, as suggested, the relevant portion in the abstract has been modified and marked (PAGE 1) to address the potential for misunderstanding.

Reviewer 2 Report

Comments and Suggestions for Authors

This manuscript by Kim et al investigated the regeneration effects of chimeric FGF21/HGFR on a murine hepatocyte cell line. Overall, it’s interesting and clearly stated. My main concerns are described as follows.

1. Specific overexpression of chimeric FGF21/HGFR in mouse liver should be explored in a mouse model after partial hepatectomy or acute liver injury.

2. For in vitro experiments, the regeneration effects of chimeric FGF21/HGFR should also be examined in a human liver cell line, which would be more supportive than in AML12 cells alone.

3. The authors stated in Results 2.1 section that they constructed stable AML12 cell line for expression of chimeric FGF21/HGFR, but the time after transfection for further analysis was not provided.

4. As seen from Fig.2C, the transfection rate in present study was not high. Although the expression of GFP could be observed by fluorescence, chimeric FGF21/HGFR expression was not confirmed by western blot, cellular immunofluorescence, or any other methods.

5. Was glycogen accumulation and LDL uptake significantly changed by FGF21/HGFR expression? Statistical analysis should be provided in Fig.2 and Fig. 3. In addition, whether there were any changes in glucose uptake should also be studied.

6. The differences in the hepatocyte survival and proliferation ability between the control group and FGF21 group should be investigated.

7. In Figure 4, the authors showed data from triplicate in each group. In view of the big variances of RT-PCR results in Fig. 4B, the authors were suggested to expand the sample sizes to provide substantial statistical power.

8. Some of the important methods and results are lacking. For example, how the cells were treated with palmitate? The RT-PCR results of both groups without palmitate treatment should be provided in Fig.4, which are vital for the confirmation of the successful establishment of NAFLD induction.

9. Although the authors evaluated the expression for hepatocyte function-related genes, the mechanisms investigation is still superficial and should be explored further. RNA seq or proteomic analyses could be considered.

10. How the chimeric transmembrane FGF21/HGFR could activate HGFR signaling and the expression of hepatocyte function-related genes should be discussed in the manuscript.

11. Some necessary references should be supplemented. For example, “… in concurrence with previous NAFLD research findings” in Discussion section.

Comments on the Quality of English Language

None

Author Response

Dear. Editors and reviewers

Thank you for your kind letter, with regard to our manuscript of ijms-2800797 together with comments. We are thankful to you for the very valuable suggestions through the whole manuscript. We tried to revise the manuscript as much as possible in line with suggestion made by the editor and reviewers. Also, we highlighted using the "Track Changes" function in Microsoft Word. We are herewith enclosing improved manuscript. Thank you again for your kind considerations.

We hope the improved version will be acceptable for publication in “IJMS”.

Thank you.

Yours sincerely,

Jun-Hyeog Jang Ph.D. (Corresponding author)

Professor

Department of Biochemistry,

Inha University School of Medicine,

Michuhol-gu, Incheon 22212, Korea

Tel: +82-32-860-9863; Fax: +82-32-885-8302. E-mail: juhjang@inha.ac.kr

Our answers to reviewer’ suggestion are as follow:

# Reviewer’s Comments

  1. Specific overexpression of chimeric FGF21/HGFR in mouse liver should be explored in a mouse model after partial hepatectomy or acute liver injury.

Answer: We appreciate the insightful suggestion for further investigation. As suggested, we added the following information in Discussion section (PAGE 6).

  1. For in vitro experiments, the regeneration effects of chimeric FGF21/HGFR should also be examined in a human liver cell line, which would be more supportive than in AML12 cells alone.

Answer: As suggested, we added the following information in Discussion section (PAGE 6).

  1. The authors stated in Results 2.1 section that they constructed stable AML12 cell line for expression of chimeric FGF21/HGFR, but the time after transfection for further analysis was not provided.

Answer: We thank the reviewer for pointing out this issue. We have clarified the time after transfection in the Methods section.

  1. As seen from Fig.2C, the transfection rate in present study was not high. Although the expression of GFP could be observed by fluorescence, chimeric FGF21/HGFR expression was not confirmed by western blot, cellular immunofluorescence, or any other methods.

Answer: Thank you for pointing out this issue. We have employed clearer images to illustrate the expression of GFP in control- or FGF21/HGFR-transfected cells.

FGF21/HGFR

Ctrl

  1. Was glycogen accumulation and LDL uptake significantly changed by FGF21/HGFR expression? Statistical analysis should be provided in Fig.2 and Fig. 3. In addition, whether there were any changes in glucose uptake should also be studied.

Answer: Thanks for your constructive suggestions. We have incorporated the quantitative data with statistics in the revised Fig. 2 and Fig. 3, showing that both glycogen accumulation and LDL uptake were significantly upregulated by FGF21/HGFR expression. Unfortunately, we haven’t examined the changes in glucose uptake levels after FGF21/HGFR transfection. In this study, we performed experiments assessing glycogen accumulation and LDL uptake to examine whether hepatocyte functions are enhanced after FGF21/HGFR transfection.

LDL uptake quantification
(Figure 3)

PAS staining quantification
(Figure 2)

  1. The differences in the hepatocyte survival and proliferation ability between the control group and FGF21 group should be investigated.

Answer: We appreciate your constructive suggestion. Although we have not performed experiments assessing hepatocyte survival and proliferation ability, there was no significant difference in cell numbers between the control and the FGF21/HGFR group at 24 h after transfection, suggesting that survival and proliferation are not influenced by FGF21/HGFR expression in the present study.

  1. In Figure 4, the authors showed data from triplicate in each group. In view of the big variances of RT-PCR results in Fig. 4B, the authors were suggested to expand the sample sizes to provide substantial statistical power.

Answer: We conducted RT-PCR multiple times, combining independent results from triplicate experiments, which may account for the observed variance. Rather than increasing the sample size, we opted for an individualized analysis, comparing the expression level of FGF21/HGFR-transfected samples to their respective controls. Despite the variations, this approach revealed statistically significant differences in the data.

  1. Some of the important methods and results are lacking. For example, how the cells were treated with palmitate? The RT-PCR results of both groups without palmitate treatment should be provided in Fig.4, which are vital for the confirmation of the successful establishment of NAFLD induction.

Answer: Thank you for pointing out these issues. We have revised the Methods section by adding information how the cells were treated with palmitate. To confirm the successful establishment of NAFLD model, we have incorporated the RT-PCR results of BSA-treated (without palmitate treatment), control group for Acc1 and Scap genes in Fig. 4A. The data shows that the levels of both genes are significantly elevated in the palmitate-treated group, indicating that NAFLD is successfully induced in hepatocytes.

  1. Although the authors evaluated the expression for hepatocyte function-related genes, the mechanisms investigation is still superficial and should be explored further. RNA seq or proteomic analyses could be considered.

Answer: Thank you for your constructive suggestion. We deeply acknowledge the need for further exploration into the mechanisms underlying FGF21/HGFR-induced hepatocyte functional enhancement. Regrettably, due to time constraints, our current study’s revision is limited. However, we are committed to conducting additional analyses and delving deeper into these mechanisms as part of our future projects.

  1. How the chimeric transmembrane FGF21/HGFR could activate HGFR signaling and the expression of hepatocyte function-related genes should be discussed in the manuscript.

Answer: As suggested, we added the following in introduction section (PAGE 2)

  1. Some necessary references should be supplemented. For example, “… in concurrence with previous NAFLD research findings” in Discussion section.

Answer: Thank you for your valuable suggestion. We made the basis of the paper more solid by including references from previous studies related to the results of this paper in the discussion sections (PAGE 5)

Round 2

Reviewer 2 Report

Comments and Suggestions for Authors

The authors have addressed most of my questions. However, the following one still needs to be dealt with seriously.

Although the expression of GFP could be observed by fluorescence in Fig.2C, the expression of functional FGF21 or HGFR should also be confirmed by other methods, such as, western blotting, and statistically compared with the controls.

In addition, there still exists some confusions related to the revised methods section. For example, in section of “4.2 Cell culture and Treatment”, the authors firstly stated that “After that, they were transfected with chimeric FGF21/HGFR DNA and control DNA for 48 h”. While then, they stated “All experiments were performed within 24 h of transfection.”

Author Response

Dear. Editors and reviewers

Thank you for your kind letter, with regard to our manuscript of ijms-2800797 together with comments. We are thankful to you for the very valuable suggestions through the whole manuscript. We tried to revise the manuscript as much as possible in line with suggestion made by the editor and reviewers. Also, we highlighted using the "Track Changes" function in Microsoft Word. We are herewith enclosing improved manuscript. Thank you again for your kind considerations.

We hope the improved version will be acceptable for publication in “IJMS”.

Thank you.

Yours sincerely,

Jun-Hyeog Jang Ph.D. (Corresponding author)

Professor

Department of Biochemistry,

Inha University School of Medicine,

Michuhol-gu, Incheon 22212, Korea

Tel: +82-32-860-9863; Fax: +82-32-885-8302. E-mail: juhjang@inha.ac.kr

Our answers to reviewer’ suggestion are as follow:

# Reviewer’s Comments

The authors have addressed most of my questions. However, the following one still needs to be dealt with seriously.

Although the expression of GFP could be observed by fluorescence in Fig.2C, the expression of functional FGF21 or HGFR should also be confirmed by other methods, such as, western blotting, and statistically compared with the controls.

Answer: Thank you for the comment. Following your suggestion, we conducted RT-PCR to confirm the expression of chimeric FGF21/HGFR in AML12 cells transfected with chimeric FGF21/HGFR DNA, comparing their expression with AML12 cells transfected with control DNA. The results showed a significant upregulation of HGFR mRNA expression in FGF21/HGFR-transfected AML12 cells compared to the controls (Fig. 2E). PAGE (3-4)

In addition, there still exists some confusions related to the revised methods section. For example, in section of “4.2 Cell culture and Treatment”, the authors firstly stated that “After that, they were transfected with chimeric FGF21/HGFR DNA and control DNA for 48 h”. While then, they stated “All experiments were performed within 24 h of transfection.”

Answer: Sorry for the confusion. We transfected the cells with FGF21/HGFR or control DAN for 48 hours and then conducted the PAS staining, LDL uptake assay, and RT-PCR using transfected cells within 24 hours after transfection. Thus, we corrected the last sentence to read, “All further assays were performed within 24 h after transfection”.

Round 3

Reviewer 2 Report

Comments and Suggestions for Authors

The authors have addressed my questions.

Author Response

The authors have addressed reviewer' questions.